# RAISIN: RESIDUAL ALGORITHMS FOR VERSATILE OFFLINE REINFORCEMENT LEARNING

**Braham Snyder**
UT Austin

**Yuke Zhu**
UT Austin

## ABSTRACT

The residual gradient algorithm (RG), gradient descent of the Mean Squared Bellman Error, brings robust convergence guarantees to bootstrapped value estimation. Meanwhile, the far more common semi-gradient algorithm (SG) suffers from well-known instabilities and divergence. Unfortunately, RG often converges slowly in practice. Baird (1995) proposed residual algorithms (RA), weighted averaging of RG and SG, to combine RG's robust convergence and SG's speed. RA works moderately well in the online setting. We find, however, that RA works disproportionately well in the offline setting. Concretely, we find that merely adding a variable residual component to SAC gives state-of-the-art scores for about half of the D4RL gym tasks. We further show that using the minimum of ten critics lets our algorithm approximately match SAC-$N$'s state-of-the-art returns using $50\times$ less compute. In contrast, TD3+BC with the same minimum-of-ten-critics trick does not match SAC-$N$'s returns on many environments. The only hyperparameter we tune is our residual weight — we leave all other hyperparameters unchanged from SAC-$N$. [1]

## 1 INTRODUCTION

Strong data scaling has given us baffling success in supervised learning. Offline reinforcement learning (offline RL) holds promise for RL to scale with that same success, among other benefits. Despite all the compelling motivations of offline RL (Levine et al., 2020), we still lack a simple, versatile, and computationally efficient solution. By versatile, we mean algorithms that attain high returns when trained on any of a diverse range of datasets, such as data collected by greatly differing policies.

Arguably the simplest and most versatile approach thus far is SAC-$N$ (An et al., 2021), which uses the minimum of $N$ critics instead of SAC's usual two critics. SAC-$N$ achieves state-of-the-art scores but, unfortunately, requires up to 500 critics for sufficient pessimism on benchmark problems. Hu et al. (2022) illustrates that stronger pessimism, specifically a smaller discount factor, enables SAC-$N$ to solve harder tasks (Rajeswaran et al., 2017). A smaller discount factor is simple and computationally efficient pessimism but not versatile — it increases bias (Zhang et al., 2020).

In this paper, we identify residual algorithms (RA) (Baird, 1995) as a simple, versatile, and computationally efficient source of pessimism for SAC-$N$. As we explained, RA saw moderate success in its goal of fusing RG's convergence with SG's speed. Recently, Zhang et al. (2019) found similar success when extending RA to deep learning. But we find RA truly excels in the *offline* setting. Prior works in both the online and offline settings (Geist et al., 2016; Fujimoto et al., 2022; Saleh & Jiang, 2019) show that, while RG performs well with data near the optimal policy, RG consistently fails when the data is far from the optimal policy. Our key insight is that RA allows for the adjustable exploitation of RG's natural pessimism. In other words, a weighted RG component may serve as a superior alternative to the widespread use of a weighted behavior cloning component for offline RL (Fujimoto & Gu, 2021; Buckman et al., 2020). Critically, however, we also find that no single weight for the RG component universally works well: you must tune it per dataset, similar to SAC-$N$ (An et al., 2021). We discuss potential routes for automatic tuning in Section 5.

We propose Raisin, roughly **RA** for **S**AC-**N**, giving D4RL (Fu et al., 2020) gym scores roughly matching SAC-$N$ — the state-of-the-art — with one-fiftieth of the critics. EDAC (An et al., 2021)

---

[1]This paper is also under review at ICLR 2023.

matches those scores as well but requires both five times more compute than Raisin and adjustment of two hyperparameters per dataset. Raisin keeps the number of critics small and fixed and solely requires adjusting one hyperparameter for pessimism (the residual weight).

Raisin easily runs at the same speed with one critic as it does with its standard ten critics (on one GPU) thanks to embarrassing parallelization, similar to SAC-$N$ (An et al., 2021). We plan to release a clean and efficient implementation of Raisin (and SAC-$N$) upon acceptance of this manuscript.

Meanwhile, TD3+BC (Fujimoto & Gu, 2021) equipped with the same minimum-of-$N$-critics tool (TD3+BC-$N$) does not appear versatile. It still does not match the scores of SAC-$N$ on a few datasets (especially the random datasets) despite tuning the pessimism per dataset. Outside of fundamental offline RL research, such simple combinations of behavior cloning and reinforcement learning are debatably the most common approach in the literature (Humphreys et al., 2022; Baker et al., 2022; Nakano et al., 2021).

IQL is computationally efficient and potentially versatile but needs more testing on suboptimal data (e.g., the random datasets of D4RL gym tasks). Maybe even more importantly, IQL is not simple. For example, IQL uses a learning rate decay for its actor, rescales its rewards by 1000, clips advantages to 100, and adds a third learning rate. Each one of these components may be overfitting. Conversely, Raisin makes a single change to SAC-$N$, and SAC-$N$ itself makes one change to SAC (Haarnoja et al., 2018), an extensively well-tested algorithm.

Upside-down reinforcement learning (UDRL) (Schmidhuber, 2019; Kumar et al., 2019) has shown some potential in recent work (Chen et al., 2021a; Lee et al., 2022), but scores poorly on suboptimal data (Brandfonbrener et al., 2022). Not to mention that it wastes capacity learning poor behaviors (Emmons et al., 2021) and that it kicks the maximization can down the road — UDRL requires a desired-return hyperparameter, a goal specification, or a complicated equivalent.

## 2 PRELIMINARIES

We consider the standard offline RL setting of a fixed dataset $\mathcal{D}$ of transitions $(s, a, r, s')$, where $s$ is a state, $a$ is an action taken in that state, $r$ is the reward received for that action, and $s'$ is the new state. We denote our policy $\pi_\phi(\cdot|s)$, whose goal is to learn actions that maximize the discounted sum of rewards $\sum_t \gamma^t r$, where $\gamma$ is the discount factor. $Q_\theta(s, a)$ is our state-action value function, whose goal is to estimate the future discounted sum of rewards given a state-action pair.

For a comprehensive overview of residual algorithms (RA), see Zhang et al. (2019). We will give a brief primer here in the context of SAC. Ignoring discounting for simplicity, SAC's loss is:

$$L(\theta_i, \mathcal{D}) := \mathop{\mathrm{E}}_{(s,a,r,s') \sim \mathcal{D}} \left[ \left( Q_{\theta_i}(s, a) - \min_{j=1,2} \bar{y}(j, r, s') \right)^2 \right],$$

where $\theta_i$ refers to the parameters $\theta$ of the $i$th of two $Q$ networks; and $\bar{y}$, the next-state target before the minimum operation, is:

$$\bar{y}(j, r, s') := r + \left( \bar{Q}_{\theta_j}(s', \tilde{a}') - \alpha \log \pi_\phi(\tilde{a}'|s') \right), \quad \tilde{a}' \sim \pi_\phi(\cdot|s').$$

$\bar{Q}$ denotes the target network (Mnih et al., 2015), and $\alpha$ is SAC's entropy coefficient.

This critic loss derives from the Mean Squared Bellman Error (MSBE), a natural error function for bootstrapped value estimation. But SAC's critic does not quite optimize the MSBE, partly because SAC, like all common value-based RL algorithms using gradient descent, ignores the gradient contribution from the next-state term. In other words, gradient descent of SAC's critic loss treats the value of the next state (plus the reward) as a fixed target towards which the current state's value is stepped. This is known as the semi-gradient (SG) algorithm. In contrast, performing true gradient descent of the MSBE is called the residual gradient (RG) algorithm. As a true gradient algorithm, RG brings robust convergence guarantees. But, thus far, RG has empirically obtained poor returns and at slow speeds. It also suffers from a few theoretical concerns (Baird, 1995; Sutton & Barto, 2018): double sampling bias (for which we discuss workarounds in Section 5); convergence to unsatisfactory

values in certain stochastic settings with function approximation; and non-identifiability (though one contribution of Patterson et al. (2021) is an identifiable MSBE). SG typically obtains higher returns (when it converges) despite its lesser convergence guarantees and instabilities. For these reasons, SG is used in practice.

In hopes of the best of both worlds, Baird (1995) proposed RA, weighted averaging of RG and SG. Zhang et al. (2019) extended RA to DDPG, introducing bidirectional target networks as an intuitive analog to the target networks commonly used for SG: since RG uses the current state and next state symmetrically for the gradient, symmetry of the current state and next state for RG's target network likely makes sense as well. Zhang et al. (2019)'s critic loss for residual DDPG with bidirectional target networks is:

$$L(\theta, \mathcal{D}) := \mathop{\mathrm{E}}_{(s,a,r,s')\sim\mathcal{D}} \left[ \left( Q_\theta(s,a) - \bar{y}(r,s') \right)^2 + \eta \left( \bar{Q}_\theta(s,a) - y(r,s') \right)^2 \right],$$

where $\eta \in [0,1]$ controls the weight for backwards bootstrapping; and $y$, unlike $\bar{y}$, uses the main $Q$ network instead of $\bar{Q}$. Both $y$ and $\bar{y}$ otherwise follow DDPG (Lillicrap et al., 2015).

## 3   RAISIN

SAC-$N$'s impressive versatility comes at the cost of massive ensembles in some environments. And as discussed in the introduction, it appears likely that even more pessimism may give SAC-$N$ yet more versatility.

Aiming to solve both challenges, we introduce Raisin, RA adapted to SAC-$N$. We incorporate Zhang et al. (2019)'s bidirectional target networks, taking the minimum of $N$ critics in the target network terms, creating the critic loss (for the $i$-th of the $N$ critics):

$$L(\theta_i, \mathcal{D}) := \mathop{\mathrm{E}}_{(s,a,r,s')\sim\mathcal{D}} \left[ \left( Q_{\theta_i}(s,a) - \min_j \bar{y}(j,r,s') \right)^2 + \eta \left( \min_j \bar{Q}_{\theta_j}(s,a) - y(i,r,s') \right)^2 \right],$$

where $j \in 1, \ldots, N$; $\eta \in [0,1]$ again controls the weight for backwards bootstrapping; and again $y$, unlike $\bar{y}$, uses the main $Q$ networks instead of the target networks $\bar{Q}$:

$$y(i,r,s') := r + \left( Q_{\theta_i}(s',\tilde{a}') - \alpha \log \pi_\phi(\tilde{a}'|s') \right), \quad \tilde{a}' \sim \pi_\phi(\cdot|s').$$

Referring to Raisin as RA + SAC-$N$ is only a rough explanation we use. Raisin is not the only possible way to combine RA and SAC-$N$. For example, we only briefly tested the alternative minimum placement $(Q - \min \bar{y})^2 + (\bar{Q} - \min y)^2$. (Where we step $\min y$ in the second term towards the average of $\bar{Q}$.) There are certainly reasons to think that formulation could be worse than Raisin (for example, it might reduce the possible range of pessimism), and Raisin indeed outscored that algorithm in preliminary experiments (not shown). That said, we don't yet consider alternative RA + SAC-$N$ approaches such as that one fully ruled out.

We have not yet considered whether $\tilde{a}'$ ought to be sampled twice, independently for $y$ and $\bar{y}$ (or independently for each critic, for that matter). In our experiments, we only sample it once, which empirically works well.

Raisin's policy update otherwise follows SAC (and SAC-$N$), maximizing only the minimum critic estimate in the policy loss. Cheng et al. (2022) observed that, with their method, such an SAC-style policy loss resulted in a limit cycle between their critics. We did not thoroughly investigate the potential for this issue in Raisin or SAC-$N$, but we did test TD3-style policy optimization (maximizing a fixed, arbitrary critic in the policy loss rather than the minimum critic) as they use, finding it not to help. We additionally tested maximizing the median, the maximum, the total average, and order statistics of the critic estimates in the policy loss, in various combinations with similar approaches to the critic loss. None worked significantly better than Raisin. This aligns with Bai et al. (2022)'s

Table 1: Score on D4RL v2 environments, mean over four seeds. For readability, scores are rounded to whole numbers, and standard deviations to two digits. Scores $\geq 90\%$ of the best score are highlighted . BC scores are taken from the SAC-$N$ paper. IQL scores were taken from its paper. SAC-$N$ and SAC-10 scores were taken from the SAC-$N$ paper where applicable. SAC-$N$ uses up to $N = 500$ critics. We use 10.

| Task Name | BC | IQL | SAC-10 ($N = 10$) | TD3+BC-10 ($N = 10$) | SAC-$N$ ($N \leq 500$) | Raisin ($N = 10$) |
|---|---|---|---|---|---|---|
| halfcheetah-random | 2 ± 0.0 | 14 | 28 ± 0.9 | 30 ± 2.1 | 28 ± 0.9 | 28 ± 0.9 |
| hopper-random | 4 ± 0.6 | 8 | 9 ± 0.7 | 17 ± 11 | 31 ± 0.0 | 31 ± 0.1 |
| walker2d-random | 1 ± 0.1 | 8 | 7 ± 12 | 4 ± 1.0 | 22 ± 0.0 | 18 ± 9.2 |
| halfcheetah-medium | 43 ± 0.6 | 47 | 68 ± 1.2 | 49 ± 2.8 | 68 ± 1.2 | 68 ± 1.2 |
| hopper-medium | 54 ± 3.8 | 66 | 4 ± 1.0 | 87 ± 10 | 100 ± 0.3 | 73 ± 21 |
| walker2d-medium | 71 ± 11 | 78 | 95 ± 1.4 | 83 ± 6.7 | 88 ± 0.2 | 95 ± 1.4 |
| halfcheetah-medium-replay | 38 ± 2.1 | 44 | 64 ± 0.8 | 44 ± 1.4 | 64 ± 0.8 | 64 ± 0.8 |
| hopper-medium-replay | 17 ± 4.8 | 95 | 102 ± 0.3 | 98 ± 12 | 102 ± 0.5 | 102 ± 0.3 |
| walker2d-medium-replay | 20 ± 9.8 | 74 | 83 ± 1.1 | 76 ± 18 | 79 ± 0.7 | 83 ± 1.1 |
| halfcheetah-medium-expert | 44 ± 1.6 | 87 | 107 ± 2.0 | 91 ± 3.7 | 107 ± 2.0 | 107 ± 2.0 |
| hopper-medium-expert | 54 ± 4.7 | 92 | 6 ± 8.0 | 113 ± 0.2 | 110 ± 0.3 | 111 ± 0.6 |
| walker2d-medium-expert | 90 ± 13 | 110 | 88 ± 58 | 110 ± 2.0 | 117 ± 0.4 | 103 ± 9.1 |
| halfcheetah-expert | 92 ± 1.5 | 94 | 105 ± 2.6 | 91 ± 0.8 | 105 ± 2.6 | 105 ± 2.6 |
| hopper-expert | 108 ± 9.7 | 110 | 1 ± 0.0 | 112 ± 0.2 | 110 ± 0.3 | 110 ± 0.4 |
| walker2d-expert | 109 ± 0.2 | 108 | 3 ± 2.8 | 110 ± 0.3 | 107 ± 2.4 | 109 ± 0.3 |

comparable conclusions regarding their policy loss. From the other direction, we ran preliminary tests of TD3+BC with an SAC-style policy loss (maximizing the minimum critic) that did not appear to improve TD3+BC.

With a fixed $\eta = 0$, and our default 10 critics, Raisin would be exactly SAC-10 (SAC-$N$ with $N = 10$). When $\eta = 0$ and $N = 2$, Raisin equates to SAC.

With a fixed $\eta = 1$, Raisin could roughly be interpreted as RG-SAC-N. We find RG-SAC-N underperforms on certain datasets, such as halfcheetah-random. This supports the conclusion of some works we discuss in Section 5, namely that RG may perform poorly when the data is far from the optimal policy. Pure RG's poor performance persisted even with extensive, combinatorial tuning, including various approaches attempting to increase optimism. We give more details in Appendix G.

We give the complete pseudocode for Raisin in Algorithm 1.

## 4 EXPERIMENTS

In all of our experiments with Raisin, we do not attempt to tune any hyperparameters other than pessimism. That is, we use the default hyperparameters of SAC-$N$, which uses the default hyperparameters of SAC aside from increasing the number of layers (for both the actor and critic) from 2 to 3, following e.g. CQL.

Like SAC-$N$ and ATAC (Cheng et al., 2022), we find that tuning the pessimism (in our case $\eta$) per dataset is crucial for performance, so we study that setting, tuning for both Raisin and the new baseline we introduce below. SAC-$N$'s experiments were tuned the same way.

### 4.1 RAISIN-10 VS. TD3+BC-10

Fixing $N = 10$, and giving TD3+BC the minimum-of-$N$-critics trick as well, we find that Raisin generally outscores TD3+BC-10 (even when we tune each method's respective pessimism on each of

Table 2: $N = 2$ scores. SAC scores from An et al. (2021). TD3+BC scores (for v2 environments) are taken from the TD3+BC appendix. Unlike our main experiments with TD3+BC-10 above, this experiment is an unfair comparison in Raisin-2's favor in that TD3+BC scores here use only 1M gradient steps and a single pessimism setting. But the TD3+BC paper's (Fujimoto & Gu, 2021) Figures 4 and 6 might imply neither difference would help TD3+BC as much as one might think.

| Task Name | TD3+BC | Raisin-2 | SAC |
|---|---|---|---|
| halfcheetah-random | 11 ± 1.1 | 30 ± 1.4 | 30 ± 1.4 |
| hopper-random | 9 ± 0.6 | 31 ± 0.2 | 10 ± 1.5 |
| walker2d-random | 2 ± 1.7 | 7 ± 7.8 | 1 ± 0.8 |
| halfcheetah-medium | 48 ± 0.3 | 63 ± 0.4 | 55 ± 28 |
| hopper-medium | 59 ± 4.2 | 71 ± 29 | 1 ± 0.0 |
| walker2d-medium | 84 ± 2.1 | 89 ± 1.0 | 0 ± 0.2 |
| halfcheetah-medium-replay | 45 ± 0.5 | 54 ± 2.0 | 1 ± 1.0 |
| hopper-medium-replay | 61 ± 19 | 102 ± 1.2 | 7 ± 0.5 |
| walker2d-medium-replay | 82 ± 5.5 | 90 ± 5.9 | 0 ± 0.3 |
| halfcheetah-medium-expert | 91 ± 4.3 | 50 ± 38 | 28 ± 19 |
| hopper-medium-expert | 98 ± 9.4 | 80 ± 57 | 1 ± 0.0 |
| walker2d-medium-expert | 110 ± 0.5 | 115 ± 1.1 | 2 ± 3.9 |
| halfcheetah-expert | 97 ± 1.1 | 43 ± 24 | -1 ± 1.8 |
| hopper-expert | 108 ± 7.0 | 31 ± 45 | 1 ± 0.0 |
| walker2d-expert | 110 ± 0.3 | 45 ± 50 | 1 ± 0.3 |

the datasets). We show this in Table 1. For example, Raisin scores 90% or higher than the best score on twelve tasks, whereas TD3+BC-10 only achieves this for seven tasks.

An et al. (2021) echoes this finding, where CQL does not match SAC-$N$'s performance even when CQL is given the $N$ critics trick, and the pessimism ($N$) is tuned per dataset. Their results with CQL-$N$ and our results with TD3+BC-10 show that the $N$ critics trick is no silver bullet for all offline RL algorithms.

As An et al. (2021) found with SAC-$N$, we find Raisin requires longer to converge on certain environments, so, like them, we run our experiments (Raisin and TD3+BC-10) for 3M gradient steps.

We give the pessimism settings for each method in Appendix A.

## 4.2 Ensemble Ablation: Raisin-2 ($N = 2$)

We find Raisin with $N = 2$, which we call Raisin-2 (approximately SAC with residual algorithms and bidirectional target networks), surprisingly improves SAC's D4RL gym scores by a median factor of 54. We show those scores in Table 2. Raisin-2 gives a competitive offline RL algorithm on its own, with negligible additional computation — but Raisin shines best with scaling (as shown in Table 1).

Along with the results of the $N = 10$ experiments, this experiment shows that the whole of Raisin is greater than the sum of its parts, and that both the RA and SAC-$N$ aspects are necessary. Raisin without SAC-$N$ performs poorly at both -expert and medium-expert datasets, and SAC-10 catastrophically fails at e.g. hopper-expert, but their careful combination matches the state of the art. The subpar scores of TD3+BC-10 show that such a combination does not always work well.

While we again use 3M gradient steps for this experiment, Raisin-2 looks like it would benefit from more gradient steps. We include learning curves in Appendices C and D.

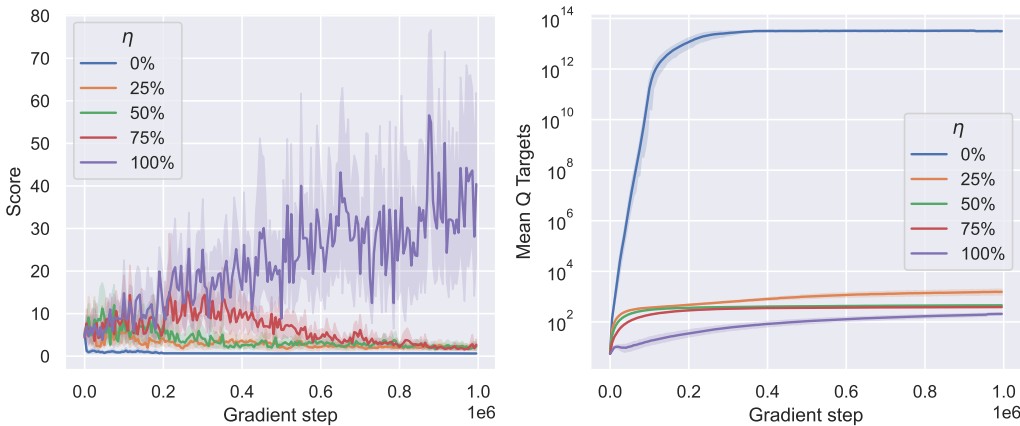

Figure 1: **Left**: Raisin-1 ($N = 1$) scores on hopper-expert with 1M gradient steps, varying $\eta$, the residual weight. $\eta = 100\%$'s massive improvement over $\eta = 0\%$ suggests our second minimum operation is not the only driver of Raisin's pessimism. **Right**: Average $Q$ targets for Raisin-1 on hopper-expert, showing that higher $\eta$ avoids unreasonably large $Q$ values. The convergence of $\eta = 0\%$ might also imply that instability alone does not account for the failure of SG.

### 4.3 WHY ARE RAS ROBUSTLY PESSIMISTIC HERE? ($N = 1$)

To our knowledge, we are the first to show that RA can induce robust pessimism, in the sense that RA can empirically turn an online algorithm into a strong offline algorithm. In these two experiments, we test two hypotheses for this robust pessimism.

**Is it the second minimum?** We first test the hypothesis that RAs are robustly pessimistic here primarily due to the second minimum operation we introduced, on the current state in the backwards bootstrapping term: $\min_j \bar{Q}_{\theta_j}(s, a)$. To do so, we run Raisin with $N = 1$, effectively removing both our new minimum operation and SAC-$N$'s original minimum operation. Even with $N = 1$, increasing the backwards bootstrapping weight $\eta$ drastically increases scores, as we show in the left plot of Figure 1.

Figure 1 (right) additionally shows that larger $\eta$ values decrease the $Q$ targets. Notably, even $\eta = 0\%$ does converge (albeit to massive values) despite poor scores, suggesting RAs indeed play an important role beyond improved convergence.

These results rule out the second minimum operation as being Raisin's primary source of increased pessimism.

**Is it value function initialization?** Wang & Ueda (2021) argue that RG (recall that RG is equivalent to RA with $\eta = 100\%$) has a tendency to maintain the average prediction. They point out that, in the tabular case, the changes in RG's value prediction at the current state and next state nearly cancel out (dependent on the discount factor) when viewing the value predictions aggregated over all states and actions. They further argue that when $Q^*$ lies above $Q$'s initialization (as it does in our experiments), RG's performance hinges on the exploration strategy chosen. This relates to the earlier observations that residual algorithms can be slow (Baird, 1995).

Accordingly, we test the effects of varying the value initialization. With $N = 1$, increasing the value initialization indeed quickly gives worse scores, shown in the left of Figure 2. And the average $Q$ targets indeed remain nearly constant over gradient steps with larger initializations, shown in the right of Figure 2.

Thus, at first sight, negative rewards, along with a standard near-zero initialization (Andrychowicz et al., 2020), might pose an issue for Raisin. However, returning to our standard $N = 10$, Raisin performs far more robustly, as we show in the left of Figure 3.

Consequently, one might expect the corresponding $Q$ targets to look like the $N = 1$ case but to just decrease faster towards $Q^*$. Instead, we see more and more of a peculiar rebound effect from

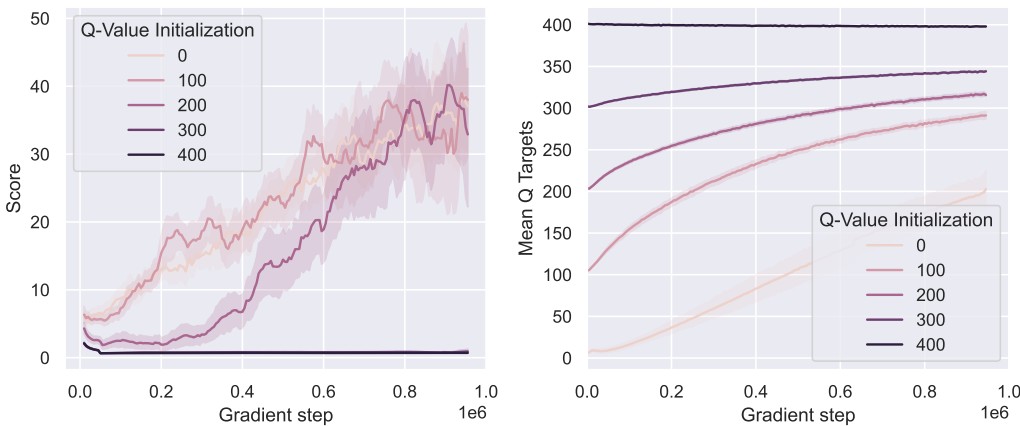

Figure 2: **Left**: Raisin-1 with $\eta = 100\%$, scores on hopper-expert when varying the value initialization. Scores deteriorate quickly as the initialization increases. For reference, Raisin-10, TD3+BC-10, and SAC-$N$ all achieve average returns of roughly 3,500 on hopper-expert (normalized scores of 110). $N = 1$ is not robust to large initializations. **Right**: Raisin-1 with $\eta = 100\%$, average $Q$ targets on hopper-expert when varying the value initialization. The slow rates of change align with Wang & Ueda (2021)'s speculation that their argument extends to the function approximation case.

excessive optimism towards excessive pessimism as we *increase* $Q$'s initialization, all despite the decent returns. We show this in the right of Figure 3.

Granted, both SAC-$N$ and Raisin sometimes exhibit similar behavior of high scores with excessively low $Q$ targets even without modifying $Q$'s initialization. For example, we reliably see this with the standard SAC-500 to solve hopper-expert (not shown).

In any case, we ran limited tests at large value initializations with re-tuning of $\eta$, further improving performance. Moreover, removing the need to tune $\eta$ may be within close reach through recent work related to RG, some of which we discuss in the following section. Not to mention, a larger $N$ nearly eliminated the issue already, and there are clear routes for additional improvements to Raisin's computational efficiency, such as DroQ (Hiraoka et al., 2021) and BatchEnsemble (Wen et al., 2020).

These results suggest value function initialization may play a key role in Raisin's pessimism. However, future work in this direction should start by examining in-depth the strange $Q$-target overshooting effect of SAC-$N$ on its own. We speculate that using independent targets might be important here, following Ghasemipour et al. (2022).

**Is it fixed points?** Xiao et al. (2021)'s findings suggest that RG and SG may converge to different fixed points in the overparameterized setting. We hypothesize this may also partly explain RG's apparent pessimism. To investigate, we train RG ($\eta = 100\%$) and SG in new runs starting from high-scoring model weights. When $N = 1$, RG and SG attain high scores on hopper-expert and halfcheetah-random, respectively. We then restart training from those high-scoring weights using both RG and SG on each environment. On hopper-expert, SG trained from RG's high-scoring weights reverts to low scores, whereas new RG runs trained from those same weights keep their high scores. Similarly, on halfcheetah-random, RG trained from SG's high-scoring weights falls to low scores, whereas new SG runs starting from those same weights keep their high scores. Our results provide evidence in favor of fixed points playing an important role. We give more details in Appendix F

## 5 RELATED WORK

### 5.1 ONLINE RESIDUAL APPROACHES

Bi-Res-DDPG (Zhang et al., 2019) reviews RG, SG, RA, and their respective fixed points. They introduce bidirectional target networks for residual algorithms in the context of DDPG. Bi-Res-DDPG shows up to a $3\times$ improvement in AUC (area under the curve of returns) over DDPG on the

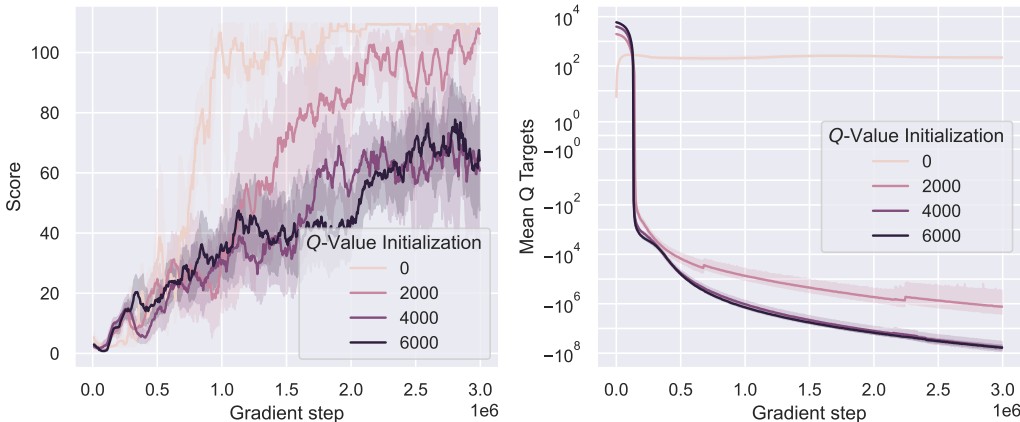

Figure 3: **Left**: Raisin-10 with $\eta = 20\%$, scores on hopper-expert when varying the value initialization. **Right**: Raisin-10 with $\eta = 20\%$, average $Q$ targets on hopper-expert when varying the value initialization. Note the strange overshooting effect at higher value initializations, even though such agents often achieve high scores. We do sometimes observe this even with the standard initialization of 0, for example, SAC-$N$ on hopper-expert with 500 critics, as suggested by An et al. (2021).

DeepMind Control Suite (Tassa et al., 2018). Whereas we identify RA as a natural fit for offline RL, propose what can be seen as Bi-Res-SAC, and show a median $50\times$ improvement in final scores over SAC. We further show that the minimum-of-$N$-critics trick synergizes with our algorithm particularly well, which lets us match SAC-$N$ with a tiny fraction of the ensemble size and outscore TD3+BC given the same $N$ critics trick. Lastly, we also delve into the sources of our robust pessimism.

C-DQN (Wang & Ueda, 2021) minimizes the maximum of RG's loss and SG's loss. Similar to RA, they claim this captures the best of both worlds, convergence and efficiency. They do not compare against RA, and they study only the online setting. A C-DQN-like approach might offer a way to avoid the need to tune $\eta$ or some equivalent. They additionally analyze potential failure modes of RG, as we discussed earlier.

## 5.2 OFFLINE RESIDUAL APPROACHES

ATAC (Cheng et al., 2022) introduces an adversarially trained algorithm with strong theoretical backing. They also incorporate RA as a fixed, equal weighting of RG and SG, using it for stability on top of their offline RL method. In contrast, we use a variable RA to control pessimism, showing among other things that RA integrated into SAC dramatically improves its returns in offline RL when $\eta$ is set appropriately.

DBRM (Saleh & Jiang, 2019) finds that RG fails at simple offline RL benchmarks when given the trajectories of a $Q$-learning agent trained online. This holds even with a fully deterministic environment (i.e., no double sampling bias, explained below), and despite the fact that $Q$-learning (a new agent trained offline on the same dataset) performs fairly well. They propose and test higher norms of the Bellman error to penalize larger residuals more, which requires more optimization steps but gets slightly better performance. It does not match SG's performance. This experimental setting is similar to D4RL's -medium-replay and -full-replay datasets, where we find intermediate values of $\eta$ to significantly outperform $\eta \in \{0, 1\}$.

Xiao et al. (2021) analyzes the fixed points of RG and SG in the overparameterized linear setting, finding they converge to different fixed points, potentially calling for increased caution around the common discussion point that overparameterized neural networks are capable of minimizing the MSBE perfectly by minimizing its projection (Fujimoto et al., 2022), the Mean Squared Projected Bellman Error (MSPBE). They further introduce two regularizers for SG and RG, again in the hopes of obtaining the best of both worlds. This is another potential avenue for replacing the need to tune $\eta$ for Raisin, or for upgrading/removing Raisin's SG component (which, as with the most commonly used RL algorithms, may cause divergence).

## 5.3 DOUBLE SAMPLING BIAS

In stochastic environments, the algorithm we have presented looks more like the naive residual gradient (naive RG) algorithm (Sutton & Barto, 2018) than the true RG. Naive RG arises when applying RG without a second, independent sample for all stochastic transitions. Naive RG, a biased estimate of true RG, no longer minimizes the MSBE. As such, naive RG can lead to poor solutions in certain stochastic environments.

Our understanding is that MuJoCo (Todorov et al., 2012) (at least in the context of the v2 environments of gym (Brockman et al., 2016), which we use in this paper) is stochastic (OpenAI, 2022; DeepMind, 2022). Although Fujimoto et al. (2022) claims otherwise, perhaps because they assume the stochasticity is minor enough to ignore. That being said, we follow Zhang et al. (2019) in ignoring this stochasticity, which empirically is not a blocker for strong performance. Granted, this suggests properly handling the double sampling bias could lead to yet stronger performance for Raisin. Many prior works, such as K-loss (Feng et al., 2019) and SBEED (Dai et al., 2017), have already tackled this issue.

## 5.4 MISCELLANEOUS RELATED WORK

**Estimation Bias and The Update-to-Data Ratio.** Offline RL and SAC-$N$, in particular, have clear ties to the literature on estimation bias in bootstrapped value learning, such as the many works making use of ensembles in ways similar to the minimum-of-$N$-critics trick, and/or high update-to-data (UTD) ratios (of which offline RL is the extreme point). This includes methods such as Maxmin $Q$-Learning (Lan et al., 2020), REDQ (Chen et al., 2021b), TQC (Kuznetsov et al., 2020), and more recently DroQ (Hiraoka et al., 2021), which have led to impressive results in real-world reinforcement learning (Smith et al., 2022).

**Theoretically Sound Algorithms.** Works such as TDC (Sutton et al., 2009), SBEED (Dai et al., 2017), and QRC (Patterson et al., 2021) use gradient corrections to build convergent online RL algorithms. Gradient correction algorithms can be seen as approximations to gradient descent of the MSBE (Patterson et al., 2021), which Raisin approximates as well (in a much different manner). We believe our empirical results may help further develop such sound algorithms, especially in the offline setting. Additionally, the adaptivity of these algorithms might point to a similar adaptive approach that obviates Raisin's need to tune $\eta$.

**Recent Criticism of The Bellman Error.** Fujimoto et al. (2022) uses both theory and experiments to point out drawbacks of the Bellman error, primarily its deficiencies concerning the value error. However, they conclude that the Bellman error can be a usable objective for on-policy evaluation. Geist et al. (2016) proposes a method for minimizing the Bellman residual through policy search (tangentially circumventing double sampling bias in the process) to more fairly compare the minimization of the Bellman residual versus direct maximization of the estimated value. They theoretically and empirically analyze the two approaches, similarly finding that their minimization of the Bellman residual performs poorly when the optimal policy is far from the data.

**Pessimism.** Buckman et al. (2020) argues that offline RL algorithms that induce pessimism by penalizing deviations from the policy that collected the data may be strictly worse than uncertainty-based approaches. This may further support the notion that Raisin and Raisin-like approaches are a more promising approach than TD3+BC and TD3+BC-like approaches.

## 6 CONCLUSION

Prior offline RL algorithms are either not versatile, not simple, or computationally inefficient. Raisin works on datasets collected by diverse behavior policies, consists of two simple changes to SAC, and requires only two percent of the compute of SAC-$N$ despite state-of-the-art scores.

Important future work includes investigating automatic alternatives to tuning $\eta$, evaluating on more challenging benchmarks, integrating a sound alternative to SG (though nearly all other RL algorithms used in practice are also based on SG), and further investigations into why Raisin works well.

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

## A  PESSIMISM SETTINGS

Table 3: TD3+BC-10's $\alpha$ multiplies $Q$ in its policy loss (higher is more optimistic). Raisin's $\eta$ multiplies its backwards bootstrapping term (lower is more optimistic).

| Task Name | TD3+BC-10: $\alpha$ | Raisin: $\eta$ ($N = 10$) | Raisin-2: $\eta$ ($N = 2$) |
|---|---|---|---|
| halfcheetah-random | 150 | 0% | 0% |
| hopper-random | 20 | 10% | 40% |
| walker2d-random | 0.5 | 1% | 20% |
| halfcheetah-medium | 4 | 0% | 20% |
| hopper-medium | 10 | 10% | 60% |
| walker2d-medium | 10 | 0% | 20% |
| halfcheetah-medium-replay | 2 | 0% | 20% |
| hopper-medium-replay | 50 | 0% | 20% |
| walker2d-medium-replay | 8 | 0% | 40% |
| halfcheetah-medium-expert | 2 | 0% | 0% |
| hopper-medium-expert | 2 | 20% | 40% |
| walker2d-medium-expert | 4 | 30% | 20% |
| halfcheetah-expert | 0.5 | 0% | 40% |
| hopper-expert | 2 | 20% | 80% |
| walker2d-expert | 1 | 5% | 60% |

## B  ALGORITHM PSEUDOCODE

---

**Algorithm 1** Raisin. Differences from SAC highlighted .

---

1: **repeat**
2:     Sample a batch $B = \{(s, a, r, s')\}$ from the offline dataset $\mathcal{D}$
3:     Compute the shared, pre-minimization, current-state targets $\bar{Q}_{\theta_j}(s, a)$ for $j \in 1, \ldots, N$.
4:     Compute the shared, pre-minimization, next-state targets for $j \in 1, \ldots, N$:

$$\bar{y}(j, r, s') := r + \left( \bar{Q}_{\theta_j}(s', \tilde{a}') - \alpha \log \pi_\phi(\tilde{a}'|s') \right), \quad \tilde{a}' \sim \pi_\phi(\cdot|s')$$

5:     Update each critic $Q_{\theta_i}$ descending its gradient:

$$\nabla_{\theta_i} \frac{1}{|B|} \sum_{(s,a,r,s') \in B} \left[ \left( Q_{\theta_i}(s, a) - \min_{j \in 1, \ldots, N} \bar{y}(j, r, s') \right)^2 + \eta \left( \min_{j \in 1, \ldots, N} \bar{Q}_{\theta_j}(s, a) - y(i, r, s') \right)^2 \right]$$

    where the next-state prediction is :

$$y(i, r, s') := r + \left( Q_{\theta_i}(s', \tilde{a}') - \alpha \log \pi_\phi(\tilde{a}'|s') \right), \quad \tilde{a}' \sim \pi_\phi(\cdot|s')$$

6:     Update the policy $\pi_\phi$ ascending its gradient:

$$\nabla_\phi \frac{1}{|B|} \sum_{s \in B} \left( \min_{j=1, \ldots, N} Q_{\phi_j}(s, \tilde{a}_\phi(s)) - \alpha \log \pi_\phi(\tilde{a}_\phi(s) \mid s) \right),$$

    where $\tilde{a}_\phi(s)$ is a sample from $\pi_\phi(\cdot \mid s)$ which is differentiable w.r.t. $\phi$ via the reparameterization trick.
7:     Update target networks with $\phi_i' \leftarrow \rho \phi_i' + (1 - \rho)\phi_i$

---

## C  RAISIN ($N = 10$) LEARNING CURVES

Like An et al. (2021), we see strange, highly non-monotonic learning over gradient steps on some datasets. This occurs in the most consistent manner on the hopper datasets, particularly hopper-random. We speculate this might be related to epoch-wise double descent (Nakkiran et al., 2019), but it's unclear. We omit a few curves where Raisin's optimal setting recovers SAC-$N$ ($\eta = 0$), since we only re-ran enough of those experiments to verify we still saw the same behavior. (Our code is built on their SAC-$N$ code.)

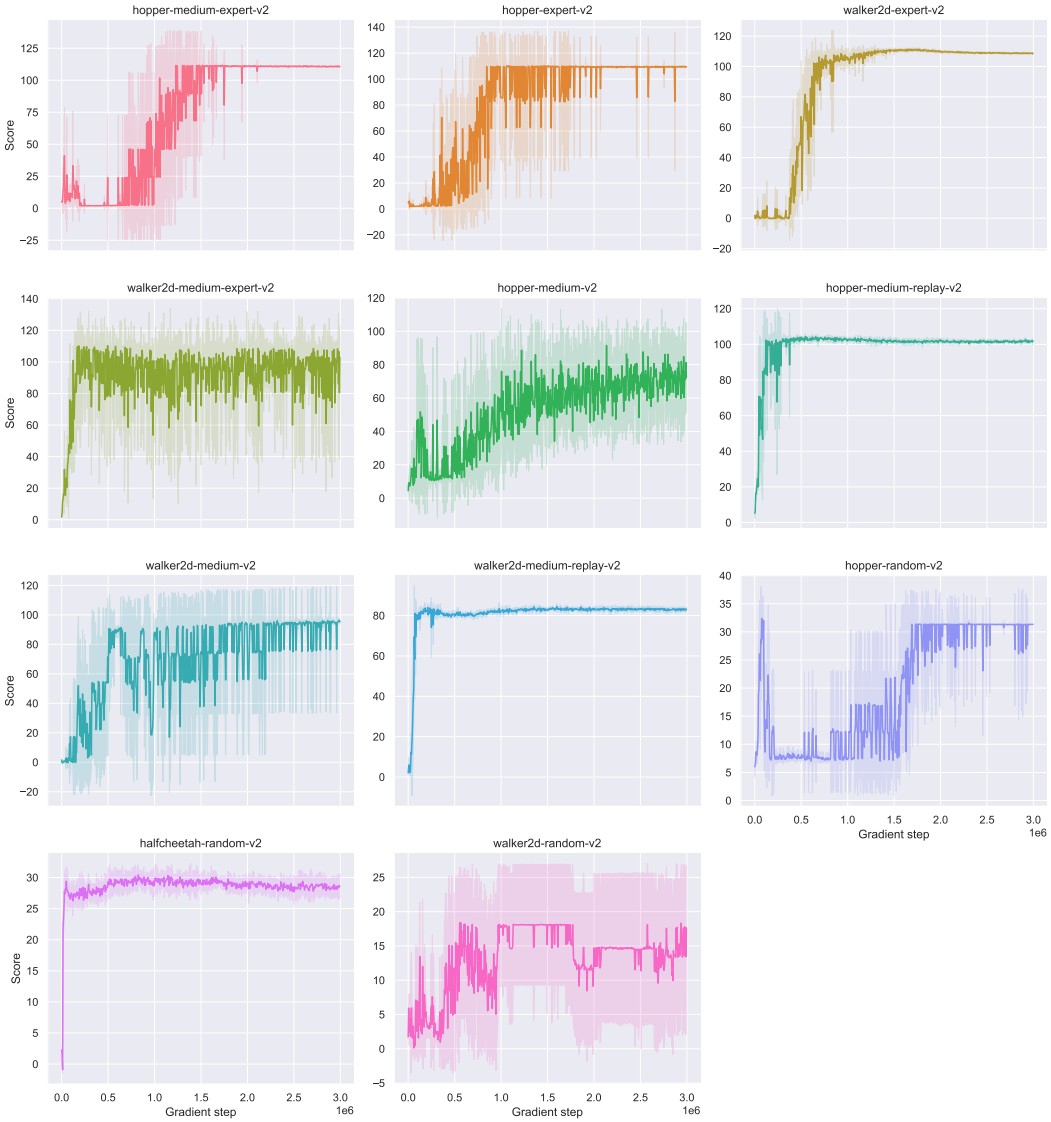

# D   RAISIN-2 ($N = 2$) LEARNING CURVES

With smaller $N$, the first peak is less sharp. For a similar depiction of this phenomenon as $N$ varies more, see An et al. (2021)'s Figure 1.

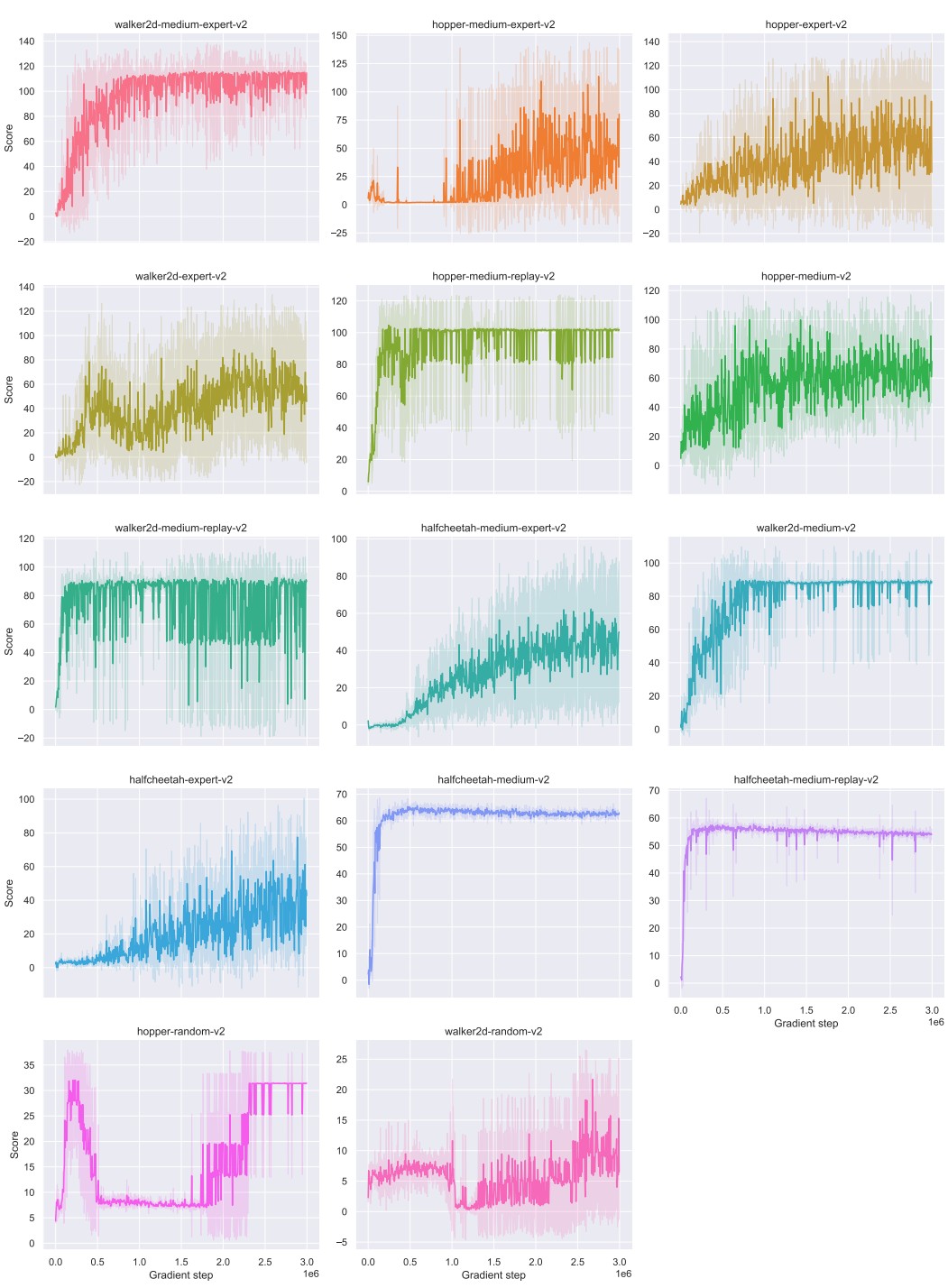

## E    A RAISIN ANALOG FOR TD3

We also test a Raisin analog for TD3 to examine the generality of our approach and to show the advantage of Raisin over a method approximately equivalent to the prior work Bi-Res-DDPG (Zhang et al., 2019) (since TD3 is essentially an improved DDPG). Our results show that RA-TD3-N (like Raisin, this is only an approximate nickname because RA + TD3 is underspecified) scores similarly well, whereas RA-TD3-1 (roughly equivalent to Bi-Res-DDPG) scores poorly. The latter is surely due to insufficient pessimism, like with Raisin-2.

We find that RA-TD3-10 does best at smaller values of $\eta$ than Raisin, with the best settings at 0.5% for hopper-random, 7.5% for walker2d-random, and 10% for hopper-expert. Additionally, we find using an SAC-style policy loss (maximizing the minimum critic instead of an arbitrary critic like TD3) performs better, so we use that here. (As we noted earlier, however, the SAC-style policy loss did not appear to help TD3+BC-10 much.) For RA-TD3-1, we find that $\eta = 100\%$ performs best, so we used that here.

Table 4: RA-TD3 scores, with Raisin scores for reference.

| Task Name | RA-TD3-1 | RA-TD3-10 | Raisin-10 |
|---|---|---|---|
| hopper-random | - | 27 ± 10 | 31 ± 0.1 |
| walker2d-random | - | 12 ± 6.3 | 18 ± 9.2 |
| hopper-expert | 12 ± 15 | 103 ± 16 | 110 ± 0.4 |

## F    FIXED POINT EXPERIMENTS

We use $N = 1$ to study the simplest case. We trained SG-SAC-1 (regular SAC, but $N = 1$) on halfcheetah-random for 250k steps, using 10 seeds. We then train both SG and RG SAC-1 (where RG-SAC-1 is $\eta = 100\%$) resuming from those weights that were trained with SG. We start from only 5 of the highest-scoring original seeds, since about half of those seeds were stuck at poor scores (because $N = 1$) and we're only interested here in how the algorithms behave once they reach a good fixed point. For each of the 5 saved SG models, we resume 2 more seeds using both SG and RG, giving 10 seeds total for RG resumes and 10 seeds total for SG resumes. We train for another 250k steps. RG's scores fall dramatically, whereas SG scores do not. We show the scores in Table 5. This experiment provides some evidence in favor of Xiao et al. (2021)'s finding that TD and BRM converge to different fixed points in the overparameterized setting. Put another way, it provides some evidence that slow convergence might not be RG's issue here (though again, Figure 2 suggests slow convergence can still be an issue in certain setups, as argued in e.g. Wang & Ueda (2021)).

Table 5: SG-SAC-1 scores on halfcheetah-random, and scores when resuming training with new runs for both RG and SG from those trained weights. SG resume scores are much higher than SG-SAC-1 scores partly because of the additional training time, and also because as discussed we resumed training only from high-scoring seeds.

| Task Name | SG-SAC-1 | RG-SAC-1 resume | SG-SAC-1 resume |
|---|---|---|---|
| halfcheetah-random | 17 ± 15 | -2.0 ± 0.3 | 32 ± 3.5 |

We also run a similar experiment training RG SAC-1 on hopper-expert for 2M steps. Again, we then train both SG and RG SAC-1 (for 500K steps) resumed from those weights. Similar to the other experiment, SG scores fall dramatically, but RG scores do not. We show these scores in Table 6. This gives additional evidence in favor of fixed points being important here.

Table 6: RG-SAC-1 scores on hopper-expert, and scores when resuming training with new runs for both RG and SG from those trained weights.

| Task Name | RG-SAC-1 | RG-SAC-1 resume | SG-SAC-1 resume |
|---|---|---|---|
| hopper-expert | 39 ± 20 | 63 ± 24 | 0.7 ± 0.1 |

We did not load the saved optimizer state — we could test loading the saved optimizer state as well, but we think this would be fairly unlikely to significantly change the results.

## G  PURE RG

We find pure RG fails to exceed an average score of 5 on halfcheetah-random, where a good score would be about 28. These poor scores persisted even with extensive, combinatorial tuning of the actor and critic learning rates, $\gamma$ (Schoknecht & Merke, 2003), the value initialization, and various ensemble aggregation techniques attempting to make RG more optimistic. For example, we simultaneously tuned each learning rate along with whether to maximize the minimum, maximum, or average of the critics in the policy loss, while at the same time tuning the order statistic of the critics to replace the minimums in the critic loss. We ran hundreds of these experiments, usually for 500k gradient steps each. None of those approaches succeeded in raising pure RG's score above 5.

