# OpenReview forum: "Raisin: Residual Algorithms for Versatile Offline Reinforcement Learning"
_NeurIPS.cc/2022/Workshop/Offline_RL — Offline RL Workshop NeurIPS 2022_

### Official Review · Reviewer_MHff · 2022-10-19

**Rating:** 4
**Confidence:** 4

**Review:**

**Summary**. This paper presents a finding that the residual gradient (RG), which takes gradient of both $(Q_{\theta} - \bar{y})^2$ and $(\bar{Q}_{\theta} - {y})^2$, performs very well for offline RL in D4RL benchmark and compatible to SAC-N though RG uses 50x less compute than SAC-N.

**Significance and novelty**.
This finding is interesting and has a potential for practical significance as it shows that a simple design can have strong empirical performance and computational benefit in offline RL setting. Simple designs with strong performance and computational benefits are desirable characteristics to make offline RL more widely adopted.

The result is however quite preliminary as the paper merely showed the performance without properly explaining why such a design works, even with empirical study. Thus, it is not clear where the performance gain actually from. An interesting case is the other extreme of RG where $\eta = \infty$ (i.e. we remove $(Q_{\theta} - \bar{y})^2$ completely from the objective and only compute gradient of $(\bar{Q}_{\theta} - {y})^2$ ). The contribution is purely empirical which does not make new algorithmic or modeling contributions. They basically plainly applied Zhang et al. (2019)’s bidirectional target networks with minimum of the critic ensemble. Though it should be fine with that almost plug-in design, to me it's more important to understand why such a design works well in offline RL (e.g. what's going in the method that makes it works that well with much less compute), than just showing the good performance. This work is promising but needs more thorough investigation as to understand the method before being accepted for representation.

---

### Official Review · Reviewer_gkh7 · 2022-10-19

**Rating:** 7
**Confidence:** 3

**Review:**

The paper introduces Raisin, an algorithm which adds a variable residual component to SAC-N. This algorithm almost matches the state of the art performance on D4RL of full SAC-N while using N=10, thereby massively reducing computational demands of the ensemble approach.

Pros: The paper is written clearly and there are given insights into how the approach could be better improved in order to attain possibly better results with this lightweight variant of SAC-N. Additionally, everything is built on top of SAC-N and hence this “skeleton” required no further tuning. In this sense, the proposal seems quite robust.

Some cons include lack of score improvement in the tasks, and failure to match desired target “ideal” performance on 3/15 tasks.